# Body Integrity Dysphoria (BID): Survey of Experts and Development of a Diagnostic Guideline

**DOI:** 10.3390/medsci13010026

**Published:** 2025-03-03

**Authors:** Erich Kasten

**Affiliations:** Practice for Behavioral Therapy and Neuropsychology, Am Krautacker 25, 23570 Travemünde, Germany; erikasten@aol.com

**Keywords:** body integrity dysphoria, body identity integrity disorder, xenomelia, amputee identity disorder, apotemnophilia

## Abstract

People who suffer from body integrity dysphoria (BID) feel a strong need to be disabled. The most common desire is for amputation or paralysis. Objectives: This study aims to gather the opinion of experts on which types of disabilities are included in BID, which therapies are useful and whether those affected should be supported in obtaining a disability. Methods: A questionnaire with 62 items and a flow chart were developed and sent to experts who have published work with regard to BID. Participants: 22 experts from 11 countries, mostly with an academic title and with an average age of 48.5 years, responded. Results: As expected, amputations and paralysis were clearly attributed to BID, other disabilities (toothlessness, incontinence, diabetes) received rather uncertain or negative scores. On average, those affected were not classified as mentally or psychiatrically ill. Neurological misconnection was considered the most likely cause. Experts did not think it was helpful to inform the health system or even the police about the desire to be disabled. Almost all experts supported the surgical solution of amputation by doctors. All participants believed that BID patients are aware of the limitations imposed by the desired disability. Finally, a flow chart is presented for diagnosis and therapy. Conclusions: The experts assume that the surgical solution is currently acceptable if it has been proven that the BID-affected person does not suffer from another mental disorder, there is a high level of suffering due to BID, other therapies have not been of any use and it is clear that the quality of life will actually increase as a result of achieving the disability.

## 1. Introduction

The greatest common denominator of body integrity dysphoria relates to the desire of a person with an intact body to want to be disabled. So far, it is not known which disabilities really fall under this category and there is uncertainty about diagnostic criteria and approaches to therapy. Despite many unanswered questions, this condition is included in the International Classification of Diseases ICD-11, as follows:

**ICD-11 6C21:** Body integrity dysphoria is characterized by an intense and persistent desire to become physically disabled in a significant way (e.g., major limb amputee, paraplegic, blind), with onset by early adolescence accompanied by persistent discomfort, or intense feelings of inappropriateness concerning current non-disabled body configuration. The desire to become physically disabled results in harmful consequences, as manifested by either the preoccupation with the desire (including time spent pretending to be disabled) significantly interfering with productivity, with leisure activities, or with social functioning (e.g., person is unwilling to have a close relationships because it would make it difficult to pretend) or by attempts to actually become disabled that result in the person putting his or her health or life in significant jeopardy [1].

Body integrity dysphoria (BID) is an extremely rare disorder; an estimation of the prevalence leads to a mean value in the range of 0.01% of the population [2]. However, many learn to live with this need and ultimately only a fraction of these seek a real operational solution, e.g., amputation of a limb. The disorder was initially referred to as “Apotemnophilia,” and later as “Amputee Identity Disorder” or “Xenomelia” [3]. The term “Body Integrity Identity Disorder” goes back to a study by Michael First [4]. Patients with BID have phases where they feel that a part of their body is not really their own. Usually, this is a leg or an arm [5]. The desire for amputation is mostly focused on the left leg, but in principle many other parts of the body can be affected [4,6]. They feel wrong with their unwanted limb and the suffering can become so great that they engage in risky behavior to remove the body part, e.g., placing the unwanted leg in dry ice or on a railway track and waiting for a train. People affected by BID are usually able to describe exactly where they want an amputation to take place. Occasionally the feeling of wrongness seems to affect the entire lower body and, in such cases, paraplegia is sought. Another way in which BID can manifest itself is the desire to lose one of the five senses, e.g., the sense of sight or hearing [7]. The most common BID variation, however, is the desire to have a limb amputated [5]. Only the disabled state is experienced as natural [8], they feel “complete” and “beautiful” if they are incomplete [9]. The intensity of the need to have a specific physical impairment increases when those affected see a person with this disability, and the desire also increases when life events or periods of overload occur [10].

The desire for a disability is experienced as an intrusive thought that is present in most everyday situations [11], and the question always hovers in the background of thoughts: How would I do that now if I were disabled? In order to make the feeling of identity with one’s own body tangible, many affected people simulate their wish. They try to create the feeling of impairment by resorting to aids such as crutches or a wheelchair. The term for using objects to simulate the disability is just “pretending” [5].

In almost all countries in the world, ethical and moral reasons currently speak against a surgical solution. Surgeons refuse to remove an intact body part, thereby creating a disability, even if the patient has suffered significantly [8]. The argument here is that the removal of an intact body part does not appear to be justified from a medical point of view; the psychological suffering is generally not taken into account.

The American psychiatrist Michael First was the first to recognize that BID can best be compared with “Gender Identity Disorder” [4,6]. As the body and the inner body feeling of those affected by BID do not match, parallels can be drawn with the identity disorder of a transsexualism/transgender/gender dysphoria [12]. Both groups, i.e., those with gender dysphoria and body integrity dysphoria, begin to notice in early childhood that “something is wrong” with their bodies. Both groups begin to playfully imitate the desired body condition, although both do it secretly because they find it embarrassing. Both groups strive in the long term to surgically adjust their real body to their mental body image. In addition, there are a number of people affected by both GD and BID who feel a strong erotic component in relation to the desired body condition. The situation becomes even more interesting when we discover that there is a very high number of patients with gender dysphoria among those affected by BID. There seems to be a common basis for this identity disorder, which is probably largely congenital and has organic and psychological causes.

In addition to the similarities that BID has with gender dysphoria, there are only few parallels with other mental illnesses that have already been classified, e.g., body dysmorphia, mania operativa or artificial syndrome.

BID is typically manifested early [4,6]. In almost all affected individuals, it occurs between the ages of four and twelve, becomes more severe during puberty and becomes more permanent with age [13]. It is striking that men are affected significantly more often than women; something which is also the case with gender dysphoria. People with BID often have an above-average level of education, they mostly work in management positions and are very successful and busy in everyday life. In addition, some overlaps can be seen in the personality structure of those affected. They are very determined, conscientious, self-confident, structured and often show narcissistic tendencies. A constant striving for challenges is also characteristic [11].

In the course of research, different explanations have emerged for the development and maintenance of body integrity dysphoria. According to Thiel [14], the desire for amputation in people with BID increases during or after a stressful situation. Amputation fantasies lead to a reduction in the unpleasant situation, which leads to operant conditioning (see also [15]). Furthermore, a developmental psychological explanation approach has also emerged, which states that a disturbance of the body schema occurs in early childhood [5]. This could also explain the onset of the desire for amputation or paralysis in early childhood. In 2022 Ho et al. found changes of pain perception between the wanted and the unwanted limb [16]. In 2020 Saetta and co-authors examined 16 patients who felt the need for a removal of the left healthy leg [17]. The primary sensorimotor area of the to-be-removed leg and the right superior parietal lobule were less functionally connected to the other brain structures. The left premotor cortex, which is involved in the multisensory integration of limb information, and the right superior parietal lobule were atrophic. The more atrophy, the stronger the desire for amputation, and the more an individual pretended to be an amputee by using wheelchairs or crutches to solve the mismatch between the desired and actual body. The findings of these authors illustrate the important role of the connectivity of different brain areas for the feeling of body ownership. They help to understand the experience of body and self as a seamless unity.

In another study, Gandola et al. [18] used fMRI to evaluate whether these findings could be replicated. These authors measured brain activations during somatosensory stimulation and motor tasks in 10 BID-patients with a need for the amputation of the left leg and 14 controls. BID individuals had reduced brain activation in the right superior parietal lobule for somatosensory stimulation and in the right paracentral lobule for the motor task. In addition, they found a reduction in the activation of somatosensory areas bilaterally. The authors conclude that BID is associated with the altered integration of somatosensory and motor signals in brain regions where the first integration of body-related signals is achieved through convergence. In 2022 Saetta et al. published another study in 16 men with BID with a long-lasting need for left leg amputation [19]. In this work these authors aimed to identify altered patterns of white matter structural connectivity. Fractional anisotropy was considered as a measure of structural connectivity. Results showed reduced structural connectivity of the right superior parietal lobule with the right cuneus, with the superior occipital and with the posterior cingulate gyri. In addition, the pars orbitalis of the right middle frontal gyrus was less connected with the putamen and the left middle temporal gyrus was less connected with the pars triangularis of the left inferior frontal gyrus. On the other hand, increased connectivity was found between the right paracentral lobule and the right caudate nucleus. These findings may consolidate the current understanding of the neural correlates of the amputation variant of BID. According to the study of Gandola et al. [18] these results show that there is a reduction in activation of somatosensory areas. These parts of the brain are regions of convergent activations for signals from the limbs and were significantly stronger in controls than in subjects with BID. Gandola et al. concluded that “BID is associated with altered integration of somatosensory and, to a lesser extent, motor signals, involving limb-specific cortical maps and brain regions where the first integration of body-related signals is achieved through convergence”.

In terms of a transversal approach, there is clearly no single cause for BID. The cause is probably a (presumably congenital) malfunction of the somatosensory system in the brain, which—similar to gender dysphoria—causes a fundamental discrepancy between the mental and real body. According to the lock and key principle, those affected react differently to the sight of disabled people even as children. This ultimately leads to a psychological component that causes emotional suffering, but on the other hand gives rise to euphoric feelings when one is in a state that comes close to the desired state.

## 2. Aim and Methods of the Study

The aim of this study was to obtain an overview of the opinions of medical and psychological experts on which symptoms belong to BID from a professional perspective and how to deal with those affected. A guideline for diagnosis was proposed and put up for discussion.

The method was an online survey with a questionnaire that included 62 items and a proposal for a flow chart for diagnosis. The questionnaire was in English and was sent by separate emails. The questionnaire mainly included items on a 7-point bipolar scale from “not at all true −3” to “+3 completely true”.

## 3. Participants

Based on a literature search of publications on body integrity dysphoria, body integrity identity disorder, xenomelia, amputee identity disorder and apotemnophilia, a total of 133 authors were identified. For 17 authors, no email address could be found or the email was returned because the address had changed. In total, two additional reminders were sent over a period of 3 months asking respondents to fill out the questionnaire. Twenty-two of the contacted experts filled out the questionnaire. The rest declined or did not respond at all. At least in one case, an expert declined to participate because she found the conceptual framework underlying the survey questionnaire to be inadequate to allow her to fully express her professional opinions.

The people who participated represented eleven countries: Australia (2), Canada (1), Czech Republic (1), Finland (1), Germany (4), Italy (4), Netherlands (1), Slovakia (1), Sweden (1), Switzerland (2) and the USA (4). Nine of the respondents had a professorial title, eleven had a doctorate, and two participants had no academic title. Twelve participants were male and ten were female. The average age was 48.5 years (24 to 70; SD 14.2). Nine had a medical education, eleven were psychologists, one was a biologist, and one was from the philosophical field. Six participants did not know anyone affected by BID personally, but had researched BID, and seven had carried out therapy with people affected by BID.

## 4. Results

### 4.1. Which Disabilities Are Included in BID?

The following values refer to a 7-point scale from −3 (rejection) to +3 (agreement).

The greatest common denominator of BID is the desire to be disabled. Scientists initially focused on amputations, and then later on the desire to be paralyzed. However, when working with BID sufferers, one also encounters strange cases, for example those who want to be deaf, hearing impaired, blind or incontinent, or who wish to suffer from diabetes. Nobody currently knows exactly where the limits of BID lie, so the experts were asked about this too.

When asked which disabilities are included in BID, the amputation of a leg or foot achieved the highest value, followed by the amputation of an arm or hand and the amputation of both legs. In fourth place is the amputation of both arms or hands, followed by the paralysis of a body part. Below this are paraplegia and visual impairment or blindness. Quadriplegia, the use of orthoses, and hearing loss/deafness counted less well, but are still in the positive area denoting what the experts would classify as BID based on the mean values. In the negative range, denoting what the experts would not classify as BID, are incontinence/wearing diapers and toothlessness. The need to suffer from diabetes is at the bottom of the list. Figure 1 and the associated table show the individual results.

### 4.2. Which Other Mental Illnesses Could BID Be Classified as?

The question of whether people with BID are psychiatrically ill was answered in the negative by the experts, but with high deviations both upwards and downwards. Figure 2 and the associated table show the results for individual mental illnesses.

### 4.3. Therapies

The next group of questions related to how the experts surveyed deal with people who suffer from the need for a disability. The questions were as follows:*Would you inform the health department or the medical officer to prevent the patient from having an amputation (or using force to achieve some other form of disability) if the patient announced their intention to do that?**Would you inform the police to stop the patient from undergoing an amputation (or using force to achieve some other form of disability)?**In your opinion, should those affected be placed in a closed psychiatric facility for their own safety?*

The results show that, on average, the experts do not consider it necessary to inform the health system. The items for involving the police or for placement in a closed ward of a psychiatric clinic are completely in the negative (see Figure 3 and the associated table).

The further list of questions then includes suggestions for therapies. The items were as follows:*Do you think antidepressants could help those affected?**Do you think neuroleptics (antipsychotics) could help those affected?**Do you think that tranquilizers could help those affected?**Do you think that psychoanalytic or depth psychological treatment could help those affected?**Do you think cognitive behavioral therapy could help those affected?**Do you think systemic therapy could help those affected?**Do you think that methods of body work, body-related therapy and relaxation techniques (e.g., autogenic training, progressive muscle relaxation, yoga) could help those affected?**Do you think BID should be treated neurologically or neuropsychologically (e.g., through transcranial magnetic stimulation)*

Drug treatment was rated negatively overall by the experts, and all psychotherapeutic procedures were also rated negatively by the experts on average (see Figure 3 and the associated table).

### 4.4. Surgical Therapies—Who Should Bear the Costs for the Disability?

The following section deals with the question of whether the surgical solution repeatedly expressed by those affected is considered sensible and who bears the costs for it, or in the long term also the costs for rehabilitation and professional reintegration. The items were as follows:*Do you think surgery that involves actual amputation (or induction of another desired disability) will help those affected?**Do you consider the amputation of a healthy part of the body (or the acquisition of another disability) due to BID to be reprehensible from a moral and/or religious point of view?**Should doctors perform amputations of a healthy body part (or the acquisition of another disability) if it is clear that the person has BID (and no other forms of mental disturbance)?**Should those affected bear the costs of an amputation (or other disability) themselves, since this procedure is carried out “at their own request”?**Should health insurance system/companies cover the full costs of such an operation?**Should those affected bear the costs for rehabilitation, physiotherapy, aids, etc. (prosthesis, wheelchair, etc.) themselves, since this procedure is carried out “at their own request”?**Should health insurance companies cover the full costs of aids (e.g., prostheses, wheelchairs, aftercare, physiotherapy, etc.)?**Should those affected have to take care of their job/occupational situation themselves once they have achieved their disability?**In your opinion, are those affected entitled to have insurance companies cover the costs of occupational reintegration after they become disabled?**Should pension insurance companies (or other institutions) finance retraining for those affected by BID who can no longer work in their old job after an amputation or other disability?**If those affected by BID are no longer able to work or are unable to work after acquiring their disability, should they then receive a disability pension even though the disability was achieved “at their own request”?*

The experts are quite clearly in favor of a surgical solution if amputation is required, and the question of whether doctors should perform such a voluntary amputation is answered in a similarly positive manner. They affirm that health care services should cover these costs (see Figure 4 and the associated table). However, several participants made a side note on this part of the question, pointing out that many countries have very different insurance systems and by no means do all of them have the legal basis to cover costs in this case. Even in countries with good insurance coverage, it will be difficult to convince institutions to pay such costs if someone voluntarily has a healthy body part amputated or voluntarily sits in a wheelchair.

### 4.5. Are Those Affected Aware of the Consequences of a Disability?

The next block of questions for the experts deals with whether those affected are really rationally aware of the consequences of creating a permanent disability. The items are as follows:*Do you think that those affected by BID are really aware of the limitations they will have to live with after an amputation (or another form of disability) once they have implemented their need?**Do you think that those affected by BID are aware in advance that phantom feelings often arise when they have implemented their need?**Do you think that those affected by BID are aware of the probability and extent that phantom pain can occur once they have implemented their need?**Do you think that those affected by BID are aware in advance of the social prejudices and stigmatization that arise when others find out that they have achieved a disability of their own volition?**Do you think that those affected by BID are really aware in advance that disabled people have greater problems finding a partner?**If BID-related interventions were carried out by doctors in your country, should those affected then have to submit psychological reports (similar to the Transsexual Act or the Gender Incongruity Guidelines)?*

With regard to limitations caused by acquiring a disability, the experts surveyed mostly assume that those affected by BID are aware of the limitations they have in their daily lives and that phantom feelings and phantom pains may occur. The participants surveyed also believe that those affected by BID are aware of the potential stigma and that it may be difficult to find a partner who accepts the disability (see Figure 5 and the associated table).

### 4.6. What Conditions Should Be Imposed?

The last part of the questions asked here examines the measures that doctors or the health system can take to ensure that someone really suffers from BID and not from another disorder. The items are as follows:*If BID-related interventions were carried out by doctors in your country, should those affected then have to submit psychiatric reports (similar to the Transsexual Act)?**If BID-related interventions were carried out by doctors in your country, should those affected then have to prove (similar to the Transsexual Law or the Gender Incongruence Guidelines) that they have tried to live in the desired disability state for at least one year—as far as this is possible?**If BID-related procedures were performed by physicians in your country, should clinics require proof of mental health?**Similar to the Transsexual Law or Gender Incongruence Guidelines, should there be a legal requirement in your country as to when and how a surgical option can be performed to induce the disabled condition?**Do you think that those affected by BID will have a permanently higher quality of life than before as a result of the desired surgical procedure?**If it were proven that those affected by BID have a permanently higher quality of life than before as a result of the desired surgical procedure, would you then support surgical treatment?*

The two transgender questions were criticized with regard to the fact that not all countries have laws for transgender people or that such operations are not performed in many countries.

The mean values of this part of the survey were exclusively positive. In concrete terms, this means that it would be considered useful if those affected had to submit a corresponding report before undergoing a surgical procedure. The experts assume that the quality of life will improve once they have reached the desired physical state (see Figure 5 and the associated table).

### 4.7. Recommendation for the Diagnostic and Treatment Process

The flow chart below was developed on the basis of previous experience in order to provide a standardization of the criteria for diagnostics and treatment (see Figure 6).

Relatively different results were obtained on some scales; as it is also interesting to know whether male experts judge differently than female experts, whether physicians judge differently than other professional groups, and whether age and personal knowledge of those affected play a role, some additional statistical procedures were calculated. Due to the small size of the sample, nonparametric methods were calculated, As no concrete hypotheses had been formulated, two-sided testing was used.

First, it was examined whether there were differences between male and female researchers. Men are more likely to deny that BID sufferers are psychiatrically ill than female experts (mean of men −1.15 ± 1.70 to woman 0.2 ± 1.60, U-test: *p* = 0.06 n.s.). Regarding the question of whether BID has a neurological cause, no significant difference was found between women (mean 0.44 ± 1.57) and men (0.77 ± 1.80, U-test: *p* = 0.53, n.s.). Although both sexes are generally in favor of amputation, men are more likely to support it than women (men 2.08 ± 0.73 to women 1.30 ± 0.90, U-test: *p* = 0.07, n.s.). This also applies to the question of whether doctors should perform an amputation (men 1.46 ± 1.22 to women 0.80 ± 1.66; U-test: *p* = 0.40 n.s.). Both groups consider it important to first provide evidence that the patient is mentally healthy—in addition to BID—but men do not consider it as important as women (men: 1.62 ± 1.60 to women 2.33±, U-test: *p* = 0.51, n.s.).

In the next analysis, we determined whether experts with a medical education (n = 9) judged differently than members of other professional groups (psychologists, biologists, philosophers, n = 13). When asked whether people with BID are psychiatrically ill, medical doctors achieved an average of 0.00 ± 0.83 and non-medical-doctors achieved an average of −1.0 ± 1.62. Thise result is not significant (U-test *p* = 0.27). With regard to neurological causes, doctors had an average of 1.56 ± 0.82 and psychologists/biologists/philosophers achieved an average of 0.00 ± 1.96. The result is not significant (U-test *p* = 0.09). Medical doctors support amputation by a mean of 2.00 ± 0.82 and the other professional groups with an average of 1.53 ± 0.93. The difference is not significant (U-test *p* = 0.33, n.s.). When asked whether surgeons should perform legal operations on people with BID, medical doctors achieved an average of 1.44 ± 1.26 and non-medicals achieved an average of 0.92 ± 1.59. The difference is not significant (U-test *p* = 0.50). When asked whether those affected should provide proof of their mental health, medical doctors on average said yes with a mean of 2.55 ± 1.09 and the other professional groups by 1.62 ± 1.50. The result is not significant (U-test *p*= 0.31).

Nearly no significant correlation was found between the age of the experts and the five selected variables. There was only a significant negative correlation between the age of the doctor or scientist and the need to provide a report proving mental health. The situation was different for the number of BID sufferers whom the experts knew personally. The more of these people the experts had actually met personally, the higher the value for supporting an amputation, including with regard to a legal operation by surgeons (see Table 1). However, there was also a significant correlation of R = 0.51 between the age of the specialist and the number of personally known BID patients.

## 5. Discussion

Although the number of people affected by BID seems to be tiny, this disorder has been included in the new version of the International Classification of Diseases (ICD-11). There are comparatively good diagnostic criteria here, but no one has yet determined what should happen to these patients if such a diagnosis is positive. We therefore need statements from professionals who have dealt with body integrity dysphoria for a future guideline in order to develop tools that provide a roadmap for how to proceed. This study makes a first small but important contribution to this.

The first aim of this study was to obtain an overview of the opinions of medical and psychological experts on which symptoms belong to BID from a professional perspective and how to deal with those affected. A guideline for diagnosis was proposed and put up for discussion. When asked which disabilities belong to BID, amputation of a leg or foot achieved the highest value, followed by amputation of an arm or hand and amputation of both legs. Amputation of both arms or hands came in fourth place, followed by paralysis of a body part. Below these are paraplegia and visual impairment or blindness. Quadriplegia, use of orthoses and hearing impairment/deafness performed less well but were still just within the range of what the experts would classify as BID based on the mean values. In the negative range of what the experts would not classify as BID are incontinence/wearing diapers and toothlessness. At the bottom of the list is the need to suffer from diabetes. The question of whether people with BID are psychiatrically ill was answered negatively by the experts, but with high deviations both upwards and downwards. The question of whether BID is delusional, could be classified as obsessive–compulsive disorder or as mania operativa, self-harming behavior, or body dysmorphic disorder (BDD) was answered negatively by the experts.

The only question that received a positive value was whether BID could be based on a neurological disorder [20]. The question of whether the cause of BID is a genetic disorder achieved an average value just below zero (neither/nor). Overprotection, neglect or abuse in childhood, secondary gain from illness, and gaining advantages from a disability are all rated negatively by the professional experts. The results show that the experts, on average, do not consider it necessary to inform the health system. The items for involving the police or for being placed in a closed ward of a psychiatric clinic are completely in the negative range. Medicinal treatment was rated rather negatively by the experts overall. Even the panacea of antidepressants received a mean value in the negative range. Tranquilizers and antipsychotics are even lower. All psychotherapeutic methods were also rated negatively by the experts. Behavioral therapy, body-oriented therapy and neuropsychological treatment are relatively close to the negative range, with psychoanalysis/depth psychology and systemic therapy even lower. The experts therefore assume that BID cannot really be cured with either medication or psychotherapy. With a positive mean value of +1.72, the experts are quite clearly in favor of a surgical solution if amputation is required, with only two participants choosing the value 0 = neither. Moral or religious concerns are not seen in the mean. The question of whether doctors should carry out such a desired amputation is answered in a similarly positive way. The doctors are opposed to the question of whether those affected should pay the costs of the operation themselves, but they agree that health care services should cover these costs. The costs of rehabilitation and aids such as prostheses, crutches or wheelchairs should also be covered by the health care system. Pensions are not available in all countries, so this question was answered very inconsistently. With regard to restrictions caused by acquiring a disability, the experts interviewed mostly assume that those affected by BID are aware of the restrictions they have in their daily lives and that phantom feelings and phantom pains may occur. The participants interviewed also believe that those affected by BID are aware of possible stigmatization and that it can be difficult to find a partner who accepts the disability.

What conditions should be required in the experts’ view to enforce the need for a disability? It was considered sensible for those affected to have to submit a psychological report before a surgical procedure; the average score for a psychiatric report was slightly lower. The requirement that proofs of mental health must be provided before surgery is about the same. Much more cautious, but still in the positive range, is the requirement that those affected by BID must have lived in a disabled state for a year before they can have an operation; this is certainly because this would be possible for some disabilities (e.g., sitting in a wheelchair while paralyzed), but not for the need for an amputation. The average vote was in favor of the need for legal regulation for the operation. The experts are very unanimous in assuming that the quality of life for people with BID improves once they have reached their desired physical state. There was also broad support for the idea that those affected by BID should be supported in order to enjoy their desired disability if it is clear that this will improve their quality of life.

There is still debate among experts about what is and is not part of BID; strangely enough, the needs of those affected focus almost exclusively on body parts such as legs or, more rarely, arms. Without having found a concrete answer, the question arises as to why, for example, ears or nose are not also affected by the desire for amputation? In this respect, it remains uncertain whether a need for blindness or deafness can be subsumed under the umbrella term BID. There is a high degree of agreement that, in addition to the need for amputation, paralysis is also part of BID, although most scientific studies deal with amputation. Nevertheless, it must be taken into account that paralysis leads to a completely different physical sensation than amputation.

There are hardly any studies on the attitude of experts to BID; in 2009, Neff and co-author surveyed experts from the medical, psychiatric and psychological fields [21]; however, most of them did not yet know what led to a correct diagnosis. Only one of the participants was willing to support the desire for amputation. Such prejudices are no longer found among the experts who have dealt with BID today, many of whom have also met those affected personally.

Regarding the causes of BID, experts are largely in agreement that early childhood trauma or secondary gain from illness is not responsible; all data point to a neurologically caused malfunction in the brain. In this respect, psychopharmacological interventions are of little use and although psychotherapy appears to be helpful in dealing with the need, it does not cure BID.

The experts’ assessment of amputation is supported by the only study to date by Noll et al. [22], which surveyed “successful” sufferers and found that all of them were satisfied or even happy with their amputation or paralysis. In this respect, almost all of the experts surveyed here support the idea that the wishes of those affected should be followed.

## Figures and Tables

**Figure 1 medsci-13-00026-f001:**
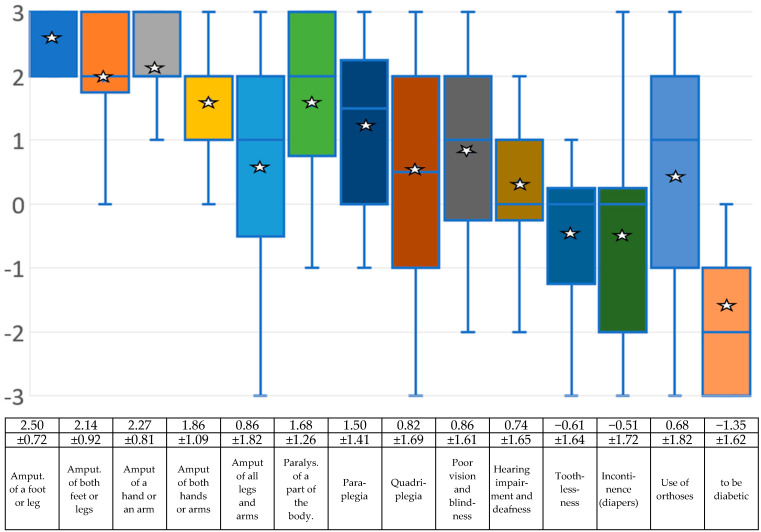
Box-and-whisker plots regarding the experts’ assessment of which disabilities belong to BID on a scale of −3 (rejection), to 0 (neither/nor), and then to +3 (agreement). The mean (white star) and standard deviation are given. Due to the small sample, the median (blue line) is also shown. The squares symbolize the upper and lower quartiles, the antennas the lowest and highest values.

**Figure 2 medsci-13-00026-f002:**
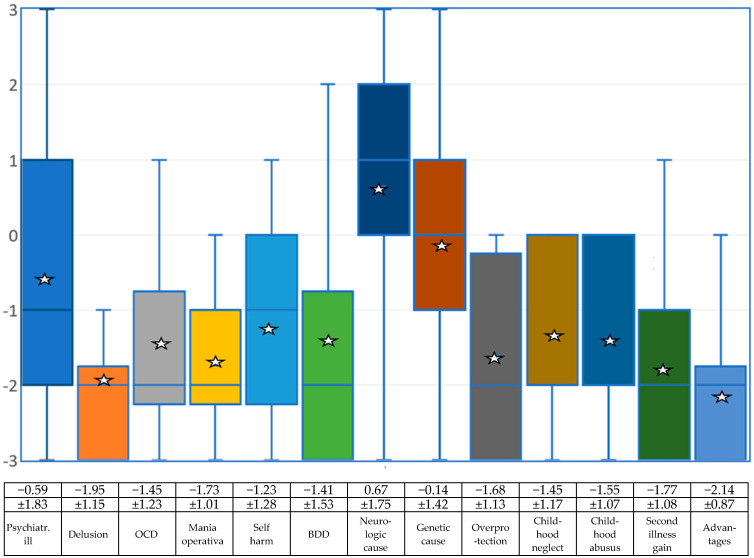
Box-and-whisker plots regarding the experts’ assessment of whether and which mental or psychiatric disorders BID could be classified as on a scale of −3 (rejection), to 0 (neither/nor) and then to +3 (agreement). The mean (star) and standard deviation are given. Due to the small sample, the median (blue line) is also shown. The squares symbolize the upper and lower quartiles, the antennas the lowest and highest values.

**Figure 3 medsci-13-00026-f003:**
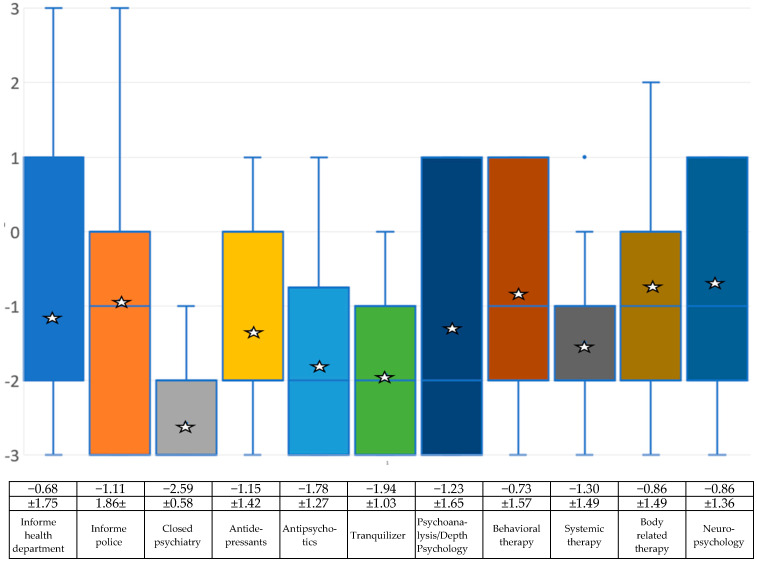
Box-and-whisker plots regarding the experts’ assessment of which drug and psychotherapeutic treatments could be used against BID on a scale of −3 (rejection), to 0 (mean) and then to +3 (approval). The mean (star) and standard deviation are given. Due to the small sample, the median (blue line) is also shown. The squares symbolize the upper and lower quartiles, the antennas the lowest and highest values.

**Figure 4 medsci-13-00026-f004:**
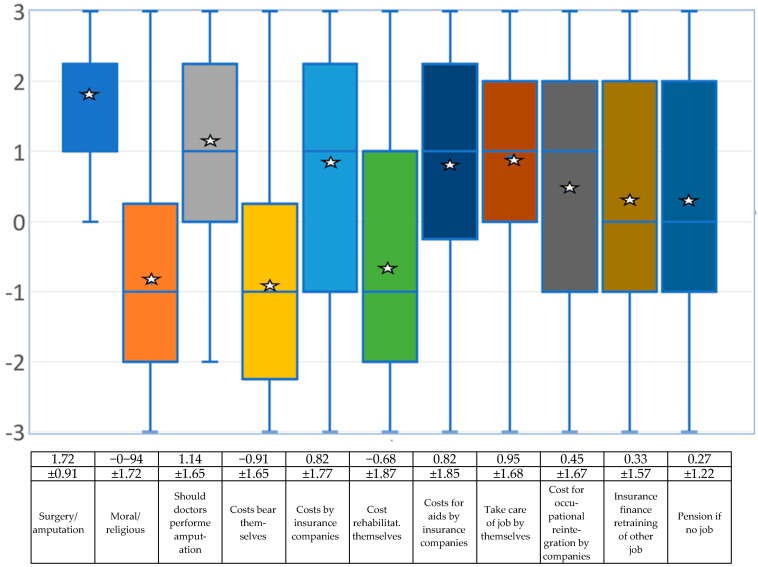
Box-and-whisker plots regarding the experts’ assessment of whether the surgical solution is of any use and who should pay the costs on a scale of −3 (rejection), to 0 (mean) and then to +3 (approval). The mean (star) and standard deviation are given. Due to the small sample, the median (blue line) is also shown. The squares symbolize the upper and lower quartiles, the antennas the lowest and highest values.

**Figure 5 medsci-13-00026-f005:**
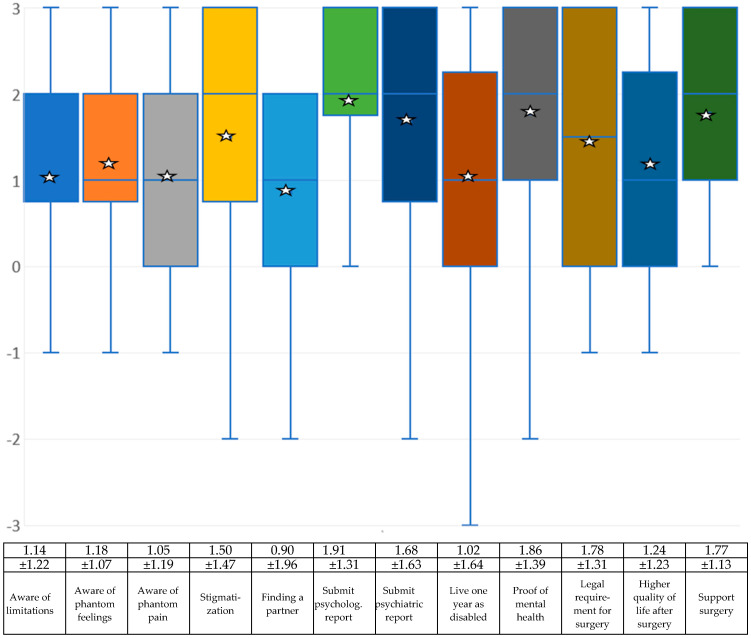
Box-and-whisker plots regarding the experts’ assessment of whether those affected are aware of the limitations of a disability, what conditions are required for an operation and whether the experts would support those affected on a scale of −3 (rejection), to 0 (mean) and then to +3 (approval). The mean (star) and standard deviation are given. Due to the small sample, the median (blue line) is also shown. The squares symbolize the upper and lower quartiles, the antennas the lowest and highest values.

**Figure 6 medsci-13-00026-f006:**
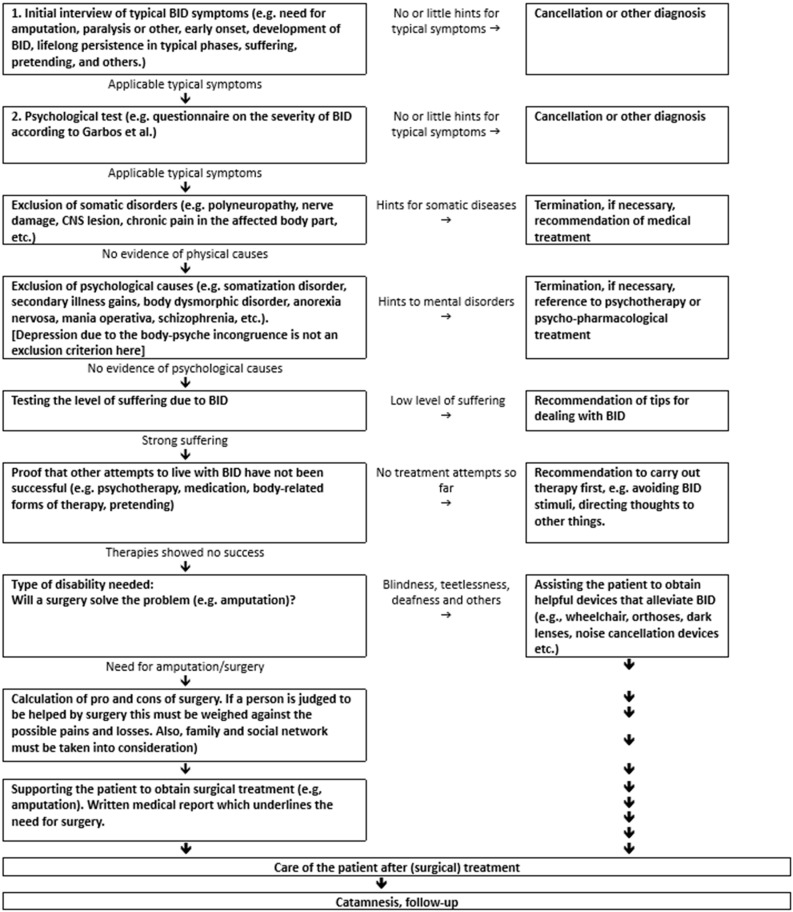
Flow chart as a suggestion for the procedure for diagnosis and treatment of patients with suspected Body Integrity Dysphoria.

**Table 1 medsci-13-00026-t001:** Correlations between age and number of personally known BID sufferers and five selected variables from the questionnaire.

	Psychiatrically Ill	Neurological Cause	Support of Amputation	Surgeons Perfom Legal Amputations	Proof of Mental Health
Age	R = −0.02 n.s.	R = 0.007 n.s.	R = 0.087 n.s.	R = 0.161 n.s.	R = −0.40 * *p* < 0.05
Number of personally met BID persons	R = −0.170 n.s.	R = −0.015 n.s.	R = 0.297 * *p* < 0.05	R = 0.322 * *p* < 0.05	R = 0.121 n.s.

*: *p* < 0.05; n.s.: not significant.

## Data Availability

The original data can be obtained from the author upon request.

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
