# Peer review of "Body Integrity Dysphoria (BID): Survey of Experts and Development of a Diagnostic Guideline"

_medsci, 2025, doi:10.3390/medsci13010026_

Round 1

Reviewer 1 Report

Comments and Suggestions for Authors

Review BID

Authors aim to gather the opinion of experts on which types of disabilities are included in BID, which therapies are useful and whether those affected should be supported in obtaining a disability. To do so, an online questionnaire was sent to experts who have published on BID.  The main results are that the experts assume that the surgical solution is currently acceptable if it has been proven that the BID affected person does not suffer from another mental disorder, there is a high level of suffering due to BID, other therapies have not been of any use and it is clear that the quality of life will actually increase as a result of achieving the disability.

The article is clear and well-written, and the goals set by the authors are new and particularly incisive in a still relatively young field and with respect to such a complex and multidisciplinary phenomenon that still requires much research.

However, we suggest a few points to improve the manuscript.

INTRODUCTION

The introduction discusses the definition of BID in great detail and mentions a historical review of the evolution of diagnosis and understanding of the neuropsychological as well as neurological and psychiatric mechanism. However, it does not elaborate on the central point, which is also the strong point of this study, namely the difficulty with which the medical world interfaces with such a transversal pathological phenomenon. Particularly around lines 75-77 it is mentioned that the aspect of the patient's psychological suffering is not sufficiently taken into account. It is suggested to elaborate further on this point, and it is suggested to insist on the need for a more transversal approach that takes into account the neurofunctional aetiology of the disorder but also considers the totality of the patient's distress.

METHODS

It is suggested that additional analyses be included to evaluate the responses of different types of authors separately. The different approach of academic figures with different backgrounds could be an important key to the data presented, in particular a comparison between the responses of specialists with a psychological background and those with a medical background. Possible convergences or divergences would allow a deeper and more realistic key to interpretation.

RESULTS

The ratio of the size of the tables and pictures makes it very difficult to understand the results. A review of the form and presentation of tables and graphs is suggested. In particular, larger tables and smaller, more aligned graphs are suggested. In addition, standard deviations for e.g. the ages of the participants should be included every time an average is reported.

DISCUSSION

Although the purpose of the article is observational, it is suggested in the discussion that the rationale of the study be taken up, i.e. to take up the underlying problem that motivated the need for this analysis. It is suggested that this aspect be explored further.

Author Response

Review BID #1

Authors aim to gather the opinion of experts on which types of disabilities are included in BID, which therapies are useful and whether those affected should be supported in obtaining a disability. To do so, an online questionnaire was sent to experts who have published on BID.  The main results are that the experts assume that the surgical solution is currently acceptable if it has been proven that the BID affected person does not suffer from another mental disorder, there is a high level of suffering due to BID, other therapies have not been of any use and it is clear that the quality of life will actually increase as a result of achieving the disability.

The article is clear and well-written, and the goals set by the authors are new and particularly incisive in a still relatively young field and with respect to such a complex and multidisciplinary phenomenon that still requires much research.

However, we suggest a few points to improve the manuscript.

INTRODUCTION

The introduction discusses the definition of BID in great detail and mentions a historical review of the evolution of diagnosis and understanding of the neuropsychological as well as neurological and psychiatric mechanism. However, it does not elaborate on the central point, which is also the strong point of this study, namely the difficulty with which the medical world interfaces with such a transversal pathological phenomenon. Particularly around lines 75-77 it is mentioned that the aspect of the patient's psychological suffering is not sufficiently taken into account. It is suggested to elaborate further on this point, and it is suggested to insist on the need for a more transversal approach that takes into account the neurofunctional aetiology of the disorder but also considers the totality of the patient's distress.

In terms of a transversal approach, there is clearly no single cause for BID. The cause is probably a (presumably congenital) malfunction of the somatosensory system in the brain, which - similar to gender dysphoria - causes a fundamental discrepancy between the mental and real body. According to the lock and key principle, those affected react differently to the sight of disabled people even as children. This ultimately leads to a psychological component that causes emotional suffering, but on the other hand gives rise to euphoric feelings when one is in a state that comes close to the desired state.

METHODS

It is suggested that additional analyses be included to evaluate the responses of different types of authors separately. The different approach of academic figures with different backgrounds could be an important key to the data presented, in particular a comparison between the responses of specialists with a psychological background and those with a medical background. Possible convergences or divergences would allow a deeper and more realistic key to interpretation.

 Relatively different results were obtained on some scales; since it is also interesting to know whether male experts judge differently than female experts, whether doctors judge differently than other professional groups, and whether age and personal knowledge of those affected play a role, some additional statistical procedures were calculated. Due to the small size of the sample, nonparametric methods were calculated, since no concrete hypotheses had been formulated, two-sided testing was used.

First, we examined whether there were differences between male and female researchers. Men are more likely to deny that BID sufferers are psychiatrically ill than female experts (mean of man -1.15 to woman 0.2, U-test: p=0.06 n.s.). Regarding the question of whether BID has a neurological cause, no significant difference was found between women (mean 0.44) and men (0.77, U-test: p=0.53, n.s.). Although both sexes are generally in favor of amputation, men are more likely to bsupport it than women (men 2.08 to women 1.30, U-test:p=0.07, n.s.). This also applies to the question of whether doctors should perform an amputation (men 1.46 to women 0.80; U-test: p=0.40 n.s.). Both groups consider it important to first provide evidence that the patient is mentally healthy - in addition to BID - but men do not consider it as important as women (men: 1.62 to women 2.33, U-test: p=0.51, n.s.).

In the next analysis, it was compared whether experts with a medical education (n=9) judged differently than members of other professional groups (psychologists, biologists, philosophers, n=13). When asked whether people with BID are psychiatrically ill, medical doctors achieved an average of 0.00 and non-medical-doctors achieved an average of -1.0. Thise result is not significant (U-test p=0.27). With regard to neurological causes, doctors had an average of 1.56 and psychologists/biologists/philosophers achieved an average of 0.20. The result is not significant (U-test p=0.09). Doctors support amputation by a mean of 2.00 and the other professional groups with an average of 1.53. The result is not significant (U-test p=0.33, n.s. ). When asked whether surgeons should perform legal operations on people with BID, doctors achieved an average of 1.44 and non-doctors achieved an average of 0.92. The result is not significant (U-test p=0.50). When asked whether those affected should provide proof of their mental health, doctors on average said yes with a mean of 2.55 and the other professional groups by 1.62. The result is not significant (U-test p= 0.31).

Tabl. 1: Correlations between age and number of personally known BID sufferers and five selected variables from the questionnaire

Psychiatrical ill

Neurological cause

Support of amputation

Surgeons perfom legal amputations

Proof of mental health

Age

R=-0.02

n.s.

R=0.007

n.s.

R=0.087

n.s.

R=0.161

n.s.

. R=-0.40*

P<0.05

Number of personally met BID-persons

R=-0.170

n.s.

R=-0.015

n.s.

R=0.297*

P<0.05

R=0.322*

P<0.05

0.121

n.s.

Nearly no significant correlation was found between the age of the experts and the five selected variables. There was only a significant negative correlation between the age of the doctor or scientist and the need to provide a report proving mental health.  The situation was different for the number of BID sufferers that the experts knew personally. The more of these people the experts had actually met personally, the higher the value for supporting an amputation, including with regard to a legal operation by surgeons (see Tab. 1). However, there was also a significant correlation of R=0.51 between the age of the specialist and the number of personally known BID patients.

RESULTS

The ratio of the size of the tables and pictures makes it very difficult to understand the results. A review of the form and presentation of tables and graphs is suggested. In particular, larger tables and smaller, more aligned graphs are suggested. In addition, standard deviations for e.g. the ages of the participants should be included every time an average is reported.

 All graphs also have an associated table that shows the standard deviation. This was actually missing for age and has been added: “The average age was 48.5 years (24 to 70; SD  14.2).“

Unfortunately, it is not entirely clear to me what I should change in the graphics? It took a lot of effort to create them and they show both the median and quartiles (which are adjusted to the sample size and data level) as well as the arithmetic mean and SD. In my opinion, the results cannot be presented any better.

DISCUSSION

Although the purpose of the article is observational, it is suggested in the discussion that the rationale of the study be taken up, i.e. to take up the underlying problem that motivated the need for this analysis. It is suggested that this aspect be explored further.

Although the number of people affected by BID seems to be tiny, this disorder has been included in the International Classification of Diseases (ICD-11). There are comparatively good diagnostic criteria here, but no one has yet determined what should happen to these patients if such a diagnosis is made. We therefore need statements from professionals who have dealt with Body Integrity Dysphoria for a future guideline in order to develop tools that provide a roadmap for how to proceed. This study makes a first small but important contribution to this.

Reviewer 2 Report

Comments and Suggestions for Authors

I found this paper quite interesting, but have a serious concern about it.  The author obtained his sample from 133 authors of articles on BID. Of these, there were only 22 people who submitted responses to the surveys (quite a small sample). It is a commonplace that people often study topics that they find interesting, perhaps because they are themselves experiencing the phenomenon they study.  Thus, the concern I have is that it may be that at least a few of the participants who responded have BID. I don’t find their responses to the first set of questions as potentially self-serving, but am concerned about some of the later responses. I would rather the author had contacted a random sample of medical doctors to learn about their responses. They could be informed about the phenomenon, and asked the same questions.

Line(s):

48: “these” should be “this”.

77: Put semi-colon after “view”, rather than a comma.

79-82: I would have liked to have a fuller discussion of the relationship between BID and GID. This is a fascinating issue.

101: Put period after “al”.

113-131: The discussion of neurological issues is fascinating, but this presentation does not “consolidate” the neurological correlates. The author needs to provide information about what the different connections may mean.

162-172: The data does not need to appear here, as it is presented thoroughly in the first figure. However, the data in the text does not always cohere with the data in the figure. Also, I don’t understand why asking these authors what is included in BID is helpful. 

229-249, Questions 4.4: I’m not sure why we are asking the authors their opinion about these issues. It would seem that whatever they think is irrelevant, as these appear to be social issues that would be decided by lawmakers, bosses, and insurance companies (whom I think will care what the experts say).

Be sure to go through all the lower parts of the figures, as there as odd dashes (e.g., “compa-nies” in Figure 4, etc.).

281-284: These should not be in italics.

302: It would never be “proven” that those with BID have a permanently higher quality of life as a result of amputation; perhaps “shown with extensive evidence” would be better.

335-337: The author begins with a statement from a book on diseases about BID. It seems to be assumed by the description that this is a mental illness.  But here we are told that the experts say that it isn’t. I am not an expert on what constitutes a mental illness, but if I were told that someone had a persistent desire to cut off a limb, I would think it was a mental illness.

359: It is difficult for me to believe that insurance companies and places of business would be willing to pay for and provide pensions for someone who voluntarily had a limb removed.

391-392: I am confused here: “someone was willing the support the desire for amputation” seems unrelated to the earlier statement, and doesn’t specify in what capacity the person was willing.

I feel great empathy for someone with these desires. It must be very difficult. But I don’t understand how finding out what 22 people who have written about the phenomenon of BID (and may themselves have BID) think about BID in relation to the questions asked is important or helpful.

Author Response

I found this paper quite interesting, but have a serious concern about it.  The author obtained his sample from 133 authors of articles on BID. Of these, there were only 22 people who submitted responses to the surveys (quite a small sample). It is a commonplace that people often study topics that they find interesting, perhaps because they are themselves experiencing the phenomenon they study.  Thus, the concern I have is that it may be that at least a few of the participants who responded have BID. I don’t find their responses to the first set of questions as potentially self-serving, but am concerned about some of the later responses. I would rather the author had contacted a random sample of medical doctors to learn about their responses. They could be informed about the phenomenon, and asked the same questions.

 It is sometimes said that scientists study what they themselves suffer from, but I am not aware of any scientific studies on this. BID is such a rare disorder that it is unlikely that any of the scientists interviewed suffer from it. Many of the participants are known to me personally, and none of them has had any ambitions to amputate a body part or to use a wheelchair in the last two decades (that is how long I have been studying BID). Unfortunately, I do not know how to deal with this criticism. You could write to all of the participants afterwards and question them, but even if they really did suffer from BID, they would not admit it, as this disorder is considered embarrassing.. All I can do is point out the statistical improbability that any of the researchers suffer from it.

Line(s):

48: “these” should be “this”. Thanks, I corrected it.

77: Put semi-colon after “view”, rather than a comma. Thanks, I corrected it.

79-82: I would have liked to have a fuller discussion of the relationship between BID and GID. This is a fascinating issue.

Both groups, i.e. those with Gender Dysphoria and Body Integrity Dysphoria, begin to notice in early childhood that something is wrong with their bodies. Both groups begin to playfully imitate the desired body condition, although both do it secretly because they find it embarrassing. Both groups strive in the long term to surgically adjust their real body to their mental body image. In addition, there are a number of people affected by both GD and BID who feel a strong erotic component in relation to the desired body condition. The situation becomes even more exciting because we find a very high number of patients with Gender Dysphoria among those affected by BID. There seems to be a common basis for this identity disorder, which is probably largely congenital and has organic and psychological causes.

101: Put period after “al”. Thanks, I corrected it.

113-131: The discussion of neurological issues is fascinating, but this presentation does not “consolidate” the neurological correlates. The author needs to provide information about what the different connections may mean.

. These findings may consolidate the current understanding of the neural correlates of the amputation variant of BID. According to the study of Gandola et al. [4] these results show that there is a reduction in activation of somatosensory areas. These parts of the brain are regions of convergent activations for signals from the limbs and were in controls significantly stronger than in subjects with BID. Gandola et al. concluded that “…BID is associated with altered integration of somatosensory and, to a lesser extent, motor signals, involving limb-specific cortical maps and brain regions where the first integration of body-related signals is achieved through convergence”.

162-172: The data does not need to appear here, as it is presented thoroughly in the first figure. However, the data in the text does not always cohere with the data in the figure. Also, I don’t understand why asking these authors what is included in BID is helpful. 

The greatest common denominator of BID is the desire to be disabled. Science initially focused on amputations, and later on the desire to be paralyzed. However, when working with BID sufferers, one also encounters strange cases, for example those who want to be deaf, hearing impaired, blind or incontinent, or who wish to suffer from diabetes. Nobody currently knows exactly where the limits of BID lie, so the experts were asked about this too.

I deleted the numbers (average and SD):

When asked which disabilities are included in BID, the amputation of a leg or foot achieved the highest value, followed by the amputation of an arm or hand and the amputation of both legs. In fourth place is the amputation of both arms or hands, followed by the paralysis of a body part. Below this are paraplegia and visual impairment or blindness. Quadriplegia, use of orthoses and hearing loss/deafness  counted less well, but are still in the positive area to what the experts would classify as BID based on the mean values. In the negative range of what the experts would not classify as BID are incontinence/wearing diapers and toothlessness. The need to suffer from diabetes is at the bottom of the list. Figure 1. and the associated table shows the individual results.

229-249, Questions 4.4: I’m not sure why we are asking the authors their opinion about these issues. It would seem that whatever they think is irrelevant, as these appear to be social issues that would be decided by lawmakers, bosses, and insurance companies (whom I think will care what the experts say).

I can understand this criticism, and the bosses probably won't really be interested in what experts think. However, I think that it is an important basis for negotiations with health insurance companies to be able to demonstrate what experts think about covering costs. I would be reluctant to delete this chapter 4.4 entirely, and the questionnaire would then be incomplete. I would therefore ask that you leave it as it is.

Be sure to go through all the lower parts of the figures, as there as odd dashes (e.g., “compa-nies” in Figure 4, etc.). Thanks, I corrected it.

281-284: These should not be in italics. Thanks, I corrected it.

302: It would never be “proven” that those with BID have a permanently higher quality of life as a result of amputation; perhaps “shown with extensive evidence” would be better.

In hindsight, the expert was absolutely right. Unfortunately, the original question in the questionnaire was "proven". I cannot change that in the article without distorting the statement in the questionnaire.

335-337: The author begins with a statement from a book on diseases about BID. It seems to be assumed by the description that this is a mental illness.  But here we are told that the experts say that it isn’t. I am not an expert on what constitutes a mental illness, but if I were told that someone had a persistent desire to cut off a limb, I would think it was a mental illness.

In terms of a transversal approach, there is clearly no single cause for BID. The cause is probably a (presumably congenital) malfunction of the somatosensory system in the brain, which - similar to gender dysphoria - causes a fundamental discrepancy between the mental and real body. According to the lock and key principle, those affected react differently to the sight of disabled people even as children. This ultimately leads to a psychological component that causes emotional suffering, but on the other hand gives rise to euphoric feelings when one is in a state that comes close to the desired state.

359: It is difficult for me to believe that insurance companies and places of business would be willing to pay for and provide pensions for someone who voluntarily had a limb removed.

However, several participants made a side note on this part of the question, pointing out that many countries have very different insurance systems and by no means all of them have the legal basis to cover costs in this case. Even in countries with good insurance coverage, it will be difficult to convince institutions to pay such costs if someone voluntarily has a healthy body part amputated or voluntarily sits in a wheelchair.

391-392: I am confused here: “someone was willing the support the desire for amputation” seems unrelated to the earlier statement, and doesn’t specify in what capacity the person was willing.

… however, most of them did not yet know what led to a correct diagnose. Only one of the participants was willing to support the desire for amputation.

I feel great empathy for someone with these desires. It must be very difficult. But I don’t understand how finding out what 22 people who have written about the phenomenon of BID (and may themselves have BID) think about BID in relation to the questions asked is important or helpful.

I am grateful that the reviewer has an understanding for those affected. Most scientists who have direct contact with BID patients also feel empathy. Ultimately, we are trying to help people who are suffering. To do this, we need guidelines for diagnosis and therapy. I think this article fulfills an important function here.

Round 2

Reviewer 1 Report

Comments and Suggestions for Authors

The authors appropriately implemented the suggestions and now the manuscript is improved significantly.

Reviewer 2 Report

Comments and Suggestions for Authors

The author has improved the manuscript by filling out the material in the introduction. The topic is fascinating.

Two minor issues:

I don't know what a "transversal approach" is (line 159). I tried looking it up, but nothing I found seems relevant. Is there a better term? Does it just mean "causal"? 

Given that there are only 22 participants, it might be useful if the author emphasized that this is a pilot study.